# Effects of Lead, Copper and Cadmium on Bioaccumulation and Translocation Factors and Biosynthesis of Photosynthetic Pigments in *Vicia faba* L. (Broad Beans) at Different Stages of Growth

Wassim Saadaoui [1,*], Hamurabi Gamboa-Rosales [2], Claudia Sifuentes-Gallardo [2], Héctor Durán-Muñoz [2,*], Khaoula Abrougui [1], Ali Mohammadi [3] and Neji Tarchoun [1]

1  High Agronomic Institute of ChottMariem, University of Sousse, Sousse 4042, Tunisia
2  Unidad Académica de Ingeniería Eléctrica, Universidad Autónoma de Zacatecas, Jardín Juarez 147, Centro Historico, Zacatecas 98000, Mexico
3  Department of Engineering and Chemical Sciences, Karlstad University, 65188 Karlstad, Sweden
*  Correspondence: wessaadaoui@gmail.com (W.S.); hectorduran3@gmail.com (H.D.-M.)

**Abstract:** Trace elements in the environmental media contribute to toxicities of different types. Their presence in the arable pedosphere is a human-health risk factor. This study focused on *Vicia faba* represented by two Tunisian varieties of bean (*Mamdouh*) and faba bean (*Badii*). The objective was to analyze the effects of lead, copper and cadmium on their growth, chlorophyll-content and carotenoids-content, as well as the bioaccumulation and translocation factor, at different stages of growth. For each metal, the concentrations the plants were subjected to were 6, 0.3 and 0.03 mg/L of the metal in the compound for lead nitrate, copper nitrate and cadmium acetate, respectively. The analysis was carried out using an atomic absorption spectrophotometer (ICP-MS), encompassing all the parts of the plant. The authors detected a perceptible decrease in the fresh weight of roots and shoots, as well as a drop in the chlorophyll and carotenoid, for all the three heavy metals. Cadmium turned out to be the most toxic of the three metals and copper (which is incidentally an essential micronutrient for plant growth) the least. As far as the bioaccumulation factor was concerned, bean and faba bean exhibited different behaviours, both with regard to the growth stages and the heavy metal absorbed. During the vegetative growth stage, both were accumulators of all the three heavy metals (a translocation factor less than unity). However, in the flowering stage, faba bean was a hyper-accumulator of copper (TF > 1); while the bean plants accumulated a lot of lead in the pods-stage (TF > 1). It is worthwhile to pose new research questions and try to answer them in this study, if legumes are accumulator or hyper accumulator plants in which stage and in where organ accumulate more HMs.

**Keywords:** bioaccumulation factor; growth stage; heavy metals; translocation factor; photosynthetic pigments; *Vicia faba*

## 1. Introduction

Heavy metals and metalloids (HMs) such as cadmium (Cd), arsenic (As), chromium (Cr), copper (Cu), mercury (Hg), nickel (Ni), lead (Pb) and zinc (Zn) are dispersed in the environmental media [1–3], and are persistent pollutants which may be transported from one environmental medium to another, or may be absorbed by the apoplast of plant roots from the pedosphere. They wend their way therefrom to the edible/non-edible parts of the plants, enter the trophic chain, and imperil human health [4,5]. Human beings are commonly exposed to Pb, Cd, Cu, As and Cr through the ingestion of contaminated food items [6,7]. For lead, cadmium and copper—the three metals of interest to the authors of this study—the Food and Agricultural Organisation and the World Health Organisation [8] have set the tolerable upper limits in food commodities as 10, 0.3 and 5 ppm respectively.

Recently-published papers have documented the occurrence of toxic HMs in both cultivated [9] and non-cultivated soils [10,11] in many regions of the world. Some heavy metals are essential micronutrients for plant growth (Cu, Fe, Zn etc.) and are indispensable for some of the metabolic processes (enzymatic activity, electron transfer, redox catalysis etc.) associated with plant growth [3,4]. On the other hand, As, Pb, Cd and Hg are detrimental to plant metabolism and are among the top 20 toxic substances, as per United States Environmental Protection Agency (USEPA) and the Agency for Toxic Substances and Disease Registry (ATSDR) [5,6].

In the arable pedosphere, the high concentrations of HMs and their subsequent bio-availability can be attributed to anthropogenic agricultural routines, which include the inefficient use of synthetic fertilizers and pesticides [7]. In many arid and semi-arid regions of the world, wastewater (partially treated) is used for irrigation, and this is a conduit for HMs to reach the soil [8]. Sewage sludge with varying concentrations of HMs is considered as a fertilizer in some parts of the world, and while being a source of essential macronutrients, it also introduces HMs to the soil. Nowadays, phytoremediation (cleaning up contaminated soils with the aid of HM-absorbing plants) is a green strategy that comprises phytoextraction, rhizo-filtration, phytodegradation, phytostabilization, and phytovolatilization for HMs in soil–food crop subsystems [9,10].

Heavy metals are significant environmental pollutants, and their toxicity is a problem of increasing significance for ecological, evolutionary, nutritional and environmental reasons. The environment is defined as the totality of circumstances surrounding an organism or group of organisms, especially the combination of external physical conditions that affect and influence the growth, development and survival of organisms [11]. A pollutant is any substance in the environment which causes objectionable effects, impairing the welfare of the environment, reducing the quality of life and which may eventually cause death. Such a substance has to be present in the environment beyond a set or tolerance limit. Hence, environmental pollution is the presence of a pollutant in the environment in air, water or soil, which may be poisonous or toxic and will cause harm to living things in the polluted environment.

The regulatory limit of cadmium (Cd) in agricultural soil is 100 mg/kg soil. Plants grown in soil containing high levels of Cd show visible symptoms of injury reflected in terms of chlorosis, growth inhibition, browning of root tips and finally death [11]. Cd has been shown to interfere with the uptake, transport and use of several elements (Ca, Mg, P and K) and water by plants [12]. Cd also reduced the absorption of nitrate and its transport from roots to shoots, by inhibiting the nitrate reductase activity in the shoots [13]. Copper (Cu) is considered as a micronutrient for plants and plays an important role in $CO_2$ assimilation and ATP synthesis. Cu is also an essential component of various proteins such as plastocyanin, of the photosynthetic system and of cytochrome oxidase of the respiratory electron transport chain [14], but an excess of Cu in soil plays a cytotoxic role, induces stress and causes injury to plants. This leads to plant growth retardation and leaf chlorosis. Exposure of plants to excess Cu generates oxidative stress and ROS [15]. Oxidative stress causes disturbance of metabolic pathways and damage to macromolecules [16].

The beneficial phytoremediative potential of plant growth-promoting rhizobacteria (PGPR) has been proven in studies conducted on cereal (rice, maize and wheat) crops by Yuan et al. [17], Nagajyoti et al. [18] and Guo et al. [19]. PGPR in the roots and rhizosphere greatly reduce heavy metal stress in plants by secreting organic acids, subsequent production of siderophores, 1-aminocyclopropane-1-carboxylic (ACC)-deaminase, phytohormones, chelation, immobilization, and enzymatic transformation [13,20]. Ramtek et al. [21] have recommended the cultivation of certain plant germoplasms possessing the ability to accumulate HMs. Generating awareness among farmers (the primary stakeholders in this regard) will be a sine qua non to incorporate and popularize this phytoremediation technology.

Leguminous crops belonging to the Fabaceae group, such as *Vicia faba* (broad beans), *Pisum sativum* (garden peas), *Phasiolus vulgaris* (common beans), and *Lens culinaris* (lentils—

white, green, red and yellow), are key components of human diets globally, as they are sources of carbohydrates, proteins, vitamins and minerals [22]. It must not be forgotten that they were among the first outputs from poor and degraded soils. The adaptability of crops from the Fabaceae group and their ability to tolerate high salinity, temperatures and even droughts, yielded different varieties in different environments (pedo-climatic), and enabled the restoration of arid ecosystems, as noted by Belimov et al. [23]. Legumes are noteworthy for their ability to capture and fix atmospheric nitrogen in the soil, through a mutualistic symbiotic relationship with soil-dwelling rhizobia bacteria. They also tend to absorb excessive amounts of HMs and accumulate them in their edible and non-edible parts [24], thus posing a clear risk to animals and humans downstream in the trophic chain [25]. Ironically, by virtue of this ability, legumes are looked upon as agents of soil-phytoremediation, when the pollutants of concern are HMs [16,26,27]. It goes without saying that whenever they are called upon to fulfill this function, they cease to be food-crops.

It has been reported that the bioaccumulation of heavy metals in vegetables, especially in leafy vegetable [28], is influenced by many factors, including climate, atmospheric deposition, soil HM concentrations, soil-type, and the degree of maturity of plants when they are harvested [29]. The concentrations of heavy metals in the plant vary according to the type of plant [30]. The ability of legumes to grow in marginal soils is often attributed to the symbiotic associations they have established with nitrogen-fixing rhizobia, and some of them tolerate extreme environmental conditions such as salinity, drought, or high temperatures. Khan et al. [31] reported that cobalt (Co), lead (Pb) and chromium (Cr) are accumulated in faba bean leaves, Fe in pods, while Zn and Cu tend to be accumulated in the seeds. While a handful of studies have been carried out on the accumulation of Pb, Cd and Cu in legumes, it is worthwhile to pose new research questions and try to answer them in this study. Therefore, the authors have decided to investigate the locations and the degrees of accumulation (bifurcated as 'accumulators' and 'hyper-accumulators') of the three HMs selected for the analysis. Such an approach may considerably upgrade all procedures aimed at selecting heavy metal-tolerant varieties to be exploited for cultivation in contaminated soil. Alternative options should be carried out in order to prevent excessive accumulation of heavy metals and all vegetables should be washed properly before consumption, as washing can remove a significant amount of aerial contamination from the vegetable surface.

In Tunisia, soil polluted with heavy metals is gradually increasing due to the scarcity of rains and the use of recycled wastewater for irrigation as is often the case in semiarid regions. In legumes, heavy metals causes various physiological and biochemical alterations and diverse toxicity symptoms such as chlorosis and necrosis [32]. Despite the acquired knowledge in relation to the response of species to heavy metals, there is a gap in relevant research fields for grain legumes. Considering that grain legumes act as accumulator plants for some heavy metals, this study aimed at investigating the response of two local varieties of grain legumes to lead, copper and cadmium at different growth stages. As such, two Tunisian varieties of common bean and faba bean were subjected to different concentrations of lead, copper and cadmium at development stage, flowering stage and pods stage, and their response was assessed on the basis of the translocation factor (TF) and the biochemical responses of the crops (chlorophyll and carotenoid contents).

## 2. Results

### 2.1. Fresh and Dry Weights of Different Parts

The effect of Pb, Cu and Cd on *Vicia faba* (common bean and faba bean) plants was evaluated by monitoring the growth of shoots and roots., based on the fresh weight of the shoots (leaves and stems) and roots (abbreviated as SFW and RFW, respectively) at different stages of growth planted in substrate (Table 1). It was seen that the growth of the roots was more sensitive to the uptake of the HMs, vis-à-vis that of the shoots (Figures 1 and 2). Among the three HMs, Cd inhibited growth the most in both shoots and roots in both common bean and faba bean, at all the different growth stages.

**Table 1.** Physico-chemical characteristics of the substrate used in this study.

| | Clay (%) | Limon (%) | Sand (%) | pH | OM (%) | Carbon (%) | Nitrogen (g/kg) | Lead (ppm) | Copper (ppm) | Cadmium (ppm) |
|---|---|---|---|---|---|---|---|---|---|---|
| Concentration | 38.7 | 27.19 | 34.54 | 7.07 | 57.33 | 33.87 | 11.31 | 0.01 | <0.15 | 0.01 |

OM: organic matter.

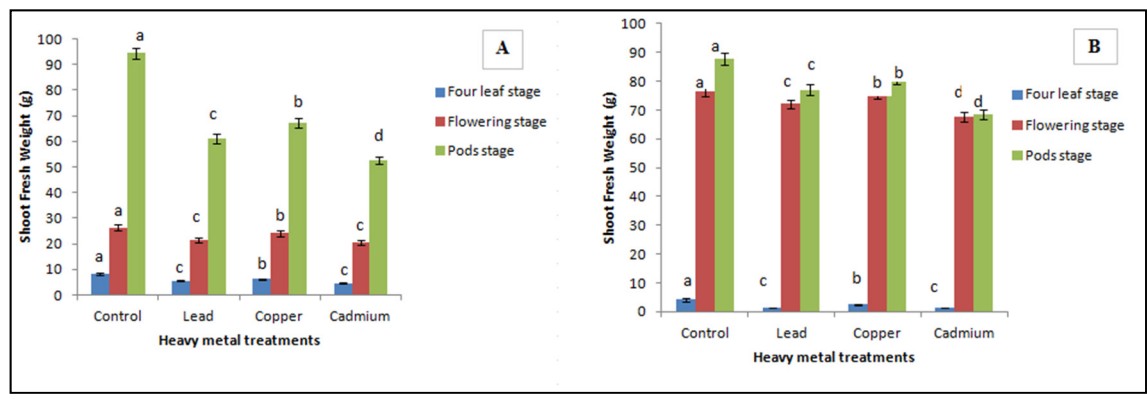

**Figure 1.** Response of bean plants (**A**) (a, b, c, d) and faba bean plants (**B**) (a, b, c, d) to HM treatments at different stages of growth, depicted for the fresh weight of the shoots (leaves and stems).

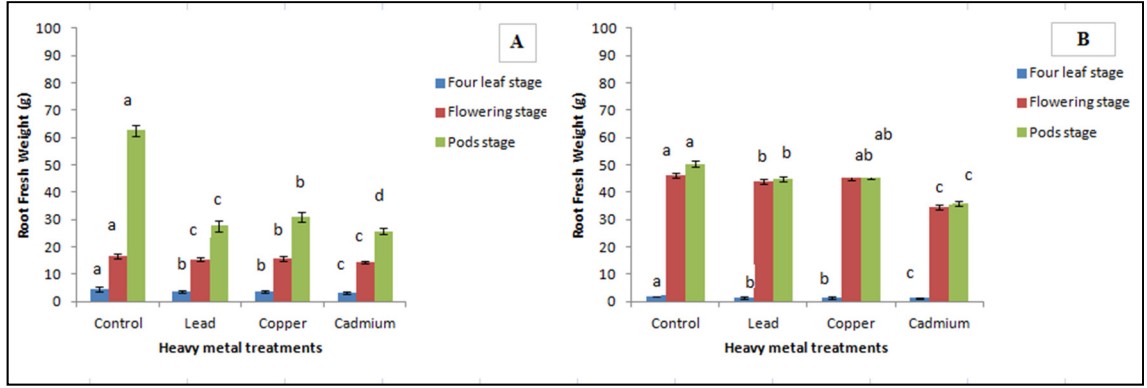

**Figure 2.** Response of bean plants (**A**) (a, b, c, d) and faba bean plants (**B**) (a, b, c) to HM treatments at different stages of growth, depicted for the fresh weight of the roots.

In the common bean samples, the SFW values were significantly reduced at all the growth stages, vis-à-vis the control plants (which were not subjected to HM-treatments). At the vegetative growth stage, the SFW varied from 4.61 g (Cd-treated plants) to 8.14 g (control plants). Plants which were exposed to Pb recorded 5.51 g (20% greater than those exposed to Cd). The decrease in the fresh weight of the shoots manifested itself in smaller sizes of the leaves for both common bean and faba bean.

During the flowering stage, the per-plant SFW values for common bean ranged between 20 and 26 g, while those for faba bean were much higher, from 67.65 to 76.63 g. The lowest in both these ranges were recorded by the plants treated with cadmium, while the highest belonged to the respective controls (Figure 1). At the stage of full-fruiting (green pods), SFW did not vary much among the treatments, though significant differences were observed vis-à-vis the respective control values. If common bean and faba bean are compared, smaller differences were observed—common bean plants had a lower number of large pods with 5–7 seeds, while faba bean plants had a higher number of small pods with 8–10 seeds per pod.

The fresh weight of roots (Figure 2) registered a much greater decrease with respect to the controls, for common bean vis-à-vis faba bean for all three HMs, with Cd affecting

the growth the most, while Cu and Pb had almost similar negative effects for both the varieties. Fresh weight of roots in vegetative, flowering and pod stages significantly decreases ($p < 0.001$) in treated plants, an effect that is more pronounced in common bean than in faba bean with respect to the control. In addition these findings show that Cd has strong negative effects on root growth indicating their sensitivity compared to the growth of shoots and fruits. These results indicate that in general the highest concentrations of metals are accumulated in roots ather than in aerial parts especially in bean plants.

### 2.2. Chlorophyll and Carotenoid Contents

Our findings indicate that all three HMs, in general, tend to decrease the total chlorophyll content of both faba bean and common bean (vis-à-vis the control), though not to the same extent. Both Cd and Pb, in that order, occasioned greater reductions, as compared to Cu. Faba bean was more resistant to copper and lead in this regard, relative to common bean. The decrease in chlorophyll content for faba bean plants treated with Cu, compared to control, was just under 15%, while that for common bean was well over 60%. The decreases registered in the carotenoid content were much less than those for chlorophyll. Quite similar to the effect on chlorophyll, Cd had the most deleterious impact and Cu the least. Here, it must be restated that Cu is an essential micronutrient and it is actually needed by plants in small quantities, while the other two are classified as toxic to plants. Regarding the carotenoid content, bean and faba bean showed the most varied response in carotenoid content to the three metals treatments (Table 2. Compared to the control, a gradual decrease of carotenoid content was observed in both varieties. The lowest values were recorded in faba bean plants both in the control and in the treated plants. Comparing the effects of the three metals on carotenoid content, cadmium had a strong negative effects compared to the control; a reduction of 59.2 and 67.51% was noted in faba bean and bean plants, respectively. In bean and faba bean plants, the chlorophylls biosynthesis was affected.

**Table 2.** Mean effect (mean $\pm$ SD) of HMs on total chlorophyll and carotenoids evaluated at the full vegetative stage of bean and faba bean plants.

| Varieties | Heavy Metals | Chlt (mg/g FW) | Car. (mg/g FW) |
|---|---|---|---|
| Common bean | Control | 15.79 [a,*] $\pm$ 0.66 | 9.05 [a] $\pm$ 1.35 |
| | Lead | 6.08 [c] $\pm$ 1.18 | 6.33 [b] $\pm$ 1.28 |
| | Copper | 6.43 [b] $\pm$ 1.12 | 6.65 [b] $\pm$ 1.14 |
| | Cadmium | 5.13 [d] $\pm$ 0.26 | 5.85 [c] $\pm$ 1.57 |
| Faba bean | Control | 13.90 [a] $\pm$ 0.19 | 4.95 [a] $\pm$ 0.25 |
| | Lead | 9.27 [c] $\pm$ 0.41 | 3.15 [c] $\pm$ 0.24 |
| | Copper | 12.10 [b] $\pm$ 0.50 | 3.74 [b] $\pm$ 1.50 |
| | Cadmium | 5.67 [d] $\pm$ 0.74 | 2.40 [d] $\pm$ 0.16 |

* Means in the same column followed by the same letter are not significantly different at $p < 0.05$, according to Duncan's Multiple Range test; Chlt: total chlorophyll; Car.: carotenoids.

### 2.3. Roots and Shoots Response during the Vegetative Stage

The data collected for the accumulation of the HMs on roots and shoots during the vegetative stage showed that the HMs tended to accumulate a lot more in the roots than in the shoots. There is a significant difference between the amounts of Cu accumulated both in roots and shoots on the one hand (between 13.5 and 14 ppm), and Pb and Cd on the other, as seen from Table 3. Pb tends to show a preference for the roots, while the accumulation of Cu and Cd in the roots is only slightly greater than that on the shoots.

**Table 3.** Root and shoot concentration of bean and faba bean at the vegetative stage.

| Species | Treatment | RC (ppm) | SC (ppm) |
|---------|-----------|----------|----------|
| Common bean | Control | 0.15 [d] ± 0.05 | 0.03 [d] ± 0.004 |
| | Lead | 4.24 [b] ± 0.30 | 1.58 [b] ± 0.13 |
| | Copper | 14.56 [a] ± 1.35 | 14.12 [a] ± 1.20 |
| | Cadmium | 0.78 [c] ± 0.08 | 0.57 [c] ± 0.04 |
| Faba bean | Control | 0.15 [d] ± 0.03 | 0.03 [d] ± 0.003 |
| | Lead | 2.25 [b] ± 0.15 | 1.08 [b] ± 0.07 |
| | Copper | 14.42 [a] ± 1.24 | 13.54 [a] ± 0.80 |
| | Cadmium | 0.89 [c] ± 0.06 | 0.71 [c] ± 0.07 |

RC: root concentration ([a, b, c, d]); SC: shoot concentration ([a, b, c, d]).

## 2.4. BAFs and TRs in the Vegetative State

Bioaccumulation factor or BAF is defined as a ratio of the concentration of a HM in the plants to the concentration in the substrate. In common bean, this was the highest for Cu at 0.81, followed by Pb with 0.43 and Cd with 0.12. The ranking remained the same for faba bean, though the BAF for Pb was much smaller than for common bean, at 0.15. The translocation factor (TF), as defined with the aid of the equations earlier and Equation (2) pertains to the TF at the vegetative stage of growth. Cu registered the highest TF for both common bean and faba bean, followed by Cd and Pb. The high values of Cu (0.9 and 0.94 in common bean and faba bean, respectively) indicate that it is readily translocated to the aerial parts of the plants owing to its role in plant metabolism (it is after all, a necessary micronutrient). Lead on the other hand has a tendency to accumulate in the roots below the ground in the vegetative stage of growth; and that means that the numerator of the TF-ratio is much less than the denominator, resulting in a lower value (Table 4).

**Table 4.** BAF and TF in bean and faba bean plants at the vegetative stage.

| Species | Treatment | BAF | TF |
|---------|-----------|-----|-----|
| Common bean | Lead | 0.43 [b] ± 0.004 | 0.35 [c] ± 0.003 |
| | Copper | 0.81 [a] ± 0.007 | 0.90 [a] ± 0.008 |
| | Cadmium | 0.12 [c] ± 0.003 | 0.77 [b] ± 0.006 |
| Faba bean | Lead | 0.15 [b] ± 0.0004 | 0.48 [c] ± 0.004 |
| | Copper | 0.94 [a] ± 0.008 | 0.94 [a] ± 0.10 |
| | Cadmium | 0.14 [b] ± 0.0003 | 0.86 [b] ± 0.008 |

BAF: bioaccumulation factor ([a, b, c]); TF: translocation factor ([a, b, c]).

## 2.5. Concentration on Roots, Shoots and Flowers

A significant difference of root, shoot and flower concentrations of treated plants vis-à-vis the controls was seen ($p < 0.001$). As seen in Table 5, the concentration of Cu is the highest in roots, flowers and shoots in faba bean plants. The Cd concentration does not increase in the roots relative to the control, but it does do so in the shoots and flowers. However, in the common bean, it is the lead concentrations which are the highest in the roots and the shoots.

**Table 5.** The HM concentrations in roots, shoots and flowers in common bean and faba bean at the flowering stage.

| Species | Treatment | RC (ppm) | SC (ppm) | FlC (ppm) |
|---|---|---|---|---|
| Common bean | Control | 1.31 [d] ± 0.20 | 0.05 [d] ± 0.0003 | 0.008 [c] ± 0.0004 |
| | Lead | 20.73 [a] ± 1.45 | 16.04 [a] ± 0.10 | 1.00 [b] ± 0.006 |
| | Copper | 13.13 [b] ± 0.88 | 4.35 [c] ± 0.45 | 2.51 [a] ± 0.15 |
| | Cadmium | 9.96 [c] ± 0.65 | 5.22 [b] ± 0.55 | 1.01 [b] ± 0.007 |
| Faba bean | Control | 1.31 [c] ± 0.008 | 0.05 [d] ± 0.0004 | 0.008 [d] ± 0.0003 |
| | Lead | 9.79 [b] ± 0.70 | 6.72 [b] ± 0.25 | 8.05 [b] ± 0.32 |
| | Copper | 13.09 [a] ± 0.91 | 17.99 [a] ± 1.06 | 19.38 [a] ± 1.05 |
| | Cadmium | 1.32 [c] ± 0.005 | 0.94 [c] ± 0.004 | 0.61 [c] ± 0.0005 |

RC: root concentration ([a, b, c, d]); SC: shoot concentration ([a, b, c, d]); FlC: flower concentration ([a, b, c, d]).

### 2.6. BAFs and TFs for HMs on Shoots and Flowers at the Flowering Stage

Table 6 shows Pb registering the highest BAF values for both common bean and faba bean. The ranking of Cu and Cd however, is different in the two cases, with the BAF for Cd being greater than that for Cu in common bean, and vice versa in the case of faba bean. The TF which represents the mobilization of HMs from the roots to the shoots (Equation (3)) was the highest for lead (0.8) in common bean, and for copper in faba bean (1.38). Interestingly, it was the least for lead (0.68) in faba bean, and the least for copper (0.32) in common bean. Higher TFs indicate a propensity for the HM in question to be distributed upward to the aerial parts of the plants more easily. A value greater than unity obviously shows that the concentration of the HM in the aerial parts (shoots in this case) is greater than that in the roots. This is an indicator of so-called 'hyper-accumulation'. As the roots conduct more and more HM atoms upward, 'space is freed', so to say, for higher uptake from the soil. Values for $TF_{flower}$ (Equation (4) in the last column in Table 6) are lower for common bean. If we consider faba bean and look at the values for Pb and Cu, they are greater than the corresponding values for $TF_{shoots}$. In general, faba bean seems to be able to hyper-accumulate and facilitate the upward transport of HMs, more effectively than common bean.

**Table 6.** BAFs and TFs for bean and faba bean plants at flowering stage.

| Species | Treatment | BAF | TF$_S$ | TF$_{Fl}$ |
|---|---|---|---|---|
| Common bean | Lead | 2.42 [a] ± 0.35 | 0.80 [a] ± 0.003 | 0.05 [c] ± 0.0005 |
| | Copper | 0.69 [c] ± 0.05 | 0.32 [c] ± 0.002 | 0.18 [a] ± 0.004 |
| | Cadmium | 1.75 [b] ± 0.004 | 0.52 [b] ± 0.006 | 0.09 [b] ± 0.0008 |
| Faba bean | Lead | 1.19 [a] ± 0.01 | 0.68 [c] ± 0.007 | 0.82 [b] ± 0.07 |
| | Copper | 1.08 [a] ± 0.03 | 1.38 [a] ± 0.18 | 1.48 [a] ± 0.38 |
| | Cadmium | 0.21 [b] ± 0.005 | 0.94 [b] ± 0.10 | 0.63 [c] ± 0.05 |

BAF: bioaccumulation factor ([a, b, c]); TF$_S$: shoot translocation factor ([a, b, c]); TF$_{Fl}$: flower translocation factor ([a, b, c]).

### 2.7. Concentrations at the Pods Stage

The results indicate that for common bean, roots accumulate more metals than shoots and pod for Pb; whereas for Cu, shoots accumulate more metals than roots and pods; and for Cd, pods accumulate more metals than roots and shoots. For faba bean, in the case of Pb and Cd, roots recorded the highest concentration; while copper concentrations are the highest in the pods and lowest in the shoots (Table 7). The BAF is the highest for Pb for both the varieties of plants. Pb tends to concentrate in the roots, leading to a lower TF value, for both beans and faba beans, be it with respect to the shoots or the pods. The ratios $TF_{shoots}$ and $TF_{pods}$ are the highest for Cd in the case of common bean (values over 3). In the case of faba bean, Cu has the highest TF when it comes to pods, while Cd takes the first position when it comes to shoots (Table 8). Common bean and faba bean are thus hyper-accumulators for Cd and Cu, respectively.

**Table 7.** The HM concentrations in roots, shoots and pods at the pods-stage.

| Species | Treatment | RC (ppm) | SC (ppm) | PC (ppm) |
|---|---|---|---|---|
| Common bean | Control | 0.34 [d] ± 0.02 | 0.02 [d] ± 0.0003 | 0.005 [d] ± 0.0003 |
| | Lead | 8.63 [b] ± 1.45 | 3.04 [b] ± 0.72 | 0.52 [c] ± 0.04 |
| | Copper | 9.96 [a] ± 1.56 | 13.17 [a] ± 1.86 | 9.48 [a] ± 1.51 |
| | Cadmium | 0.65 [c] ± 0.07 | 0.60 [c] ± 0.05 | 0.73 [b] ± 0.08 |
| Faba bean | Control | 0.34 [d] ± 0.03 | 0.02 [c] ± 0.0003 | 0.005 [d] ± 0.0004 |
| | Lead | 25.07 [a] ± 2.05 | 3.04 [b] ± 0.42 | 2.87 [b] ± 0.28 |
| | Copper | 14.34 [b] ± 1.18 | 5.25 [a] ± 0.57 | 27.04 [a] ± 2.19 |
| | Cadmium | 5.84 [c] ± 0.89 | 3.58 [b] ± 0.38 | 1.44 [c] ± 0.17 |

RC: root concentration ([a, b, c, d]); SC: shoot concentration ([a, b, c, d]); PC: pod concentration ([a, b, c, d]).

**Table 8.** Bioaccumulation factor and translocation factor in bean and faba bean plants at pods stage.

| Species | Treatment | BAF | TF$_S$ | TF$_P$ |
|---|---|---|---|---|
| Common bean | Lead | 1.09 [a] ± 0.05 | 0.33 [c] ± 0.02 | 0.05 [c] ± 0.0004 |
| | Copper | 0.56 [b] ± 0.006 | 1.33 [b] ± 0.15 | 0.95 [b] ± 0.05 |
| | Cadmium | 0.16 [c] ± 0.008 | 3.05 [a] ± 0.85 | 3.20 [a] ± 0.77 |
| Faba bean | Lead | 4.07 [a] ± 0.89 | 0.11 [c] ± 0.01 | 0.11 [c] ± 0.05 |
| | Copper | 1.79 [c] ± 0.34 | 0.35 [b] ± 0.02 | 1.88 [b] ± 0.14 |
| | Cadmium | 1.88 [b] ± 0.45 | 0.62 [a] ± 0.05 | 0.28 [a] ± 0.05 |

BAF: bioaccumulation factor ([a, b, c]); TF$_S$: shoot translocation factor ([a, b, c]); TF$_P$: pod translocation factor ([a, b, c]).

*2.8. Correlations among Main Variables*

Pearson correlation coefficients among the different variables are calculated and shown in Table 9 (vegetative stage) and 10 (flowering stage), using the SAS software. In the vegetative stage, a highly positive correlation (above 0.9) is seen between SC and RC, BAF and RC, and BAF and SC. The TF$_{veg}$ is poorly-correlated with SC, RC and BAF, indicating that at the vegetative stage, the HMs obviously are largely concentrated in the roots. These correlations are true for both common bean and faba bean at the vegetative stage.

**Table 9.** Pearson correlation between main variables at vegetative stage of *Vicia faba*.

| | RC | SC | Subst C | BAF | TFveg |
|---|---|---|---|---|---|
| RC | - | | | | |
| SC | 0.98 ** | - | | | |
| Subst C | 0.80 ** | 0.77 ** | - | | |
| BAF | 0.95 ** | 0.92 ** | 0.66 ** | - | |
| TF veg | 0.37 * | 0.47 ** | ns | 0.31 * | - |

RC: roots concentration, SC: shoots concentration (**), SubstC. (**): substrate concentration (**), BAF: bioaccumulation factor (**), TF$_{veg}$: translocation factor (*) at vegetative stage, ns: not significant.

However, at the flowering stage, medium, positive and negative correlations were respectively seen between the varieties (common bean or faba bean) on the one hand, and RC, FlC and BAF on the other. Nevertheless, the translocation factor for flowers has a highly-positive correlation with both the HM concentration in the flowers (0.87**) and TF$_{shoots}$ (0.78**) (Table 9). The correlation coefficients and the results tabulated in the earlier tables clearly show that common bean and faba bean behave very differently when it comes to translocation of the HMs from roots and shoots to flowers. At the pod stage (fruiting), BAF is positively and significantly correlated with variety (0.70**) and RC (0.78**). (BAF and RC (0.77**)) and (TF$_{fl}$ * Fl C (0.87**)) are two other highly-positive correlations observed (Table 10). All this leads to inference that *Vicia faba* is a hyper-accumulator of HMs at the fruiting stage (when there is a greater transfer of HMs from below the ground to the aerial parts of the plants).

**Table 10.** Pearson correlation coefficients among main variables at flowering stage of *Vicia faba*.

|  | Var. | RC | SC | Fl C | Subst C | BAF | TFshoot | TF fl |
|---|---|---|---|---|---|---|---|---|
| Var. | - |  |  |  |  |  |  |  |
| RC | −0.51 ** | - |  |  |  |  |  |  |
| SC | ns | 0.74 ** | - |  |  |  |  |  |
| Fl C | 0.57 ** | ns | 0.61 ** | - |  |  |  |  |
| Subst C | ns | ns | ns | ns | - |  |  |  |
| BAF | −0.48 ** | 0.77 ** | 0.56 ** | ns | 0.56 ** | - |  |  |
| TFshoot | 0.56 ** | ns | 0.49 ** | 0.60 ** | 0.64 ** | 0.35 * | - |  |
| TFfl | 0.82 ** | ns | ns | 0.87 ** | ns | 0.78 ** | ns | - |

Var: variety, RC: roots concentration (**), SC: shoots concentration (**), FlC: flower concentration (**), SubstC: substrate concentration, BAF: bioaccumulation factor (**), $TF_{shoots}$: translocation factor (*) at flowering stage, $TF_{fl}$: translocation factor flowers (**), ns: (not significant).

## 3. Material and Methods

### 3.1. Plant Growth Conditions

The experiment was carried out in a greenhouse at the experimental station of the High Agronomic Institute of ChottMariem (Sousse, Tunisia). Local Bean (*Vicia faba* L. cv. Mamdouh) and faba bean (*Vicia faba* L. cv. Badii) seeds were rinsed in water for 24 h and were sown in pots (one seed per pot) containing fresh earth and commercial peat (1/3 *v/v*, respectively); the latter being added to increase the organic matter content in the substrate. The physical and chemical characteristics of the substrate have been tabulated in Table 1. During the process of germination, tap water was used for drip irrigation. A thin 2.5 cm-layer of sandy soil was added to ensure water drainage and prevent water logging. A total of 600 plastic pots (volume of (3 L) were used, 90 per crop-variety and HM (90 × 2 varieties × 3 HMs = 540), while 30 pots served as controls.

### 3.2. Treatments Applied

Three weeks after the seeds were sown, HMs were added, twice per week. The concentration was 6 mg/L of the metal in the compound for lead, 0.3 mg/L for copper and 0.03 mg/L for cadmium, for lead nitrate [$Pb(NO_3)_2$], copper nitrate [$Cu(NO_3)_2$] and cadmium acetate [$(CH_3COO)_2Cd\text{-}2H_2O$] from sigma Aldrich, USA, respectively. These concentrations were applied based on the limits set by the FAO/WHO [8] while treatment with tap water served as a control. Each treatment was applied, using 3 replications. During the experiment, plants were irrigated with a nutrient solution of fertilizer (NPK: 20-20-20) twice a week. Pots were randomly moved daily to minimize any position-related effects.

### 3.3. Samples Collection

At different stages of growth (vegetative stage and reproductive stage) for each pot, four different parts of the plant (roots, stems, leaves and pods) were collected separately, weighed (fresh weight, FW), dried at 60 °C, ground and stored at ambient temperature in flasks for inductively-coupled plasma mass spectrometry (ICP-MS) analysis. The cleaning procedure adopted for the roots can be gleaned from Wang et al. [33].

### 3.4. Assay for Chlorophylls and Total Carotenoids

To measure the chlorophyll and carotenoid contents at anthesis stage, three mature leaves from each pot were selected. The sample-extraction process was guided by Curtis and Shetty (1996) and Yang et al. (1998). Fifty milligrams of leaf tissue (in triplicate) were assimilated into 3 mL of methanol, and stored at 23 °C in darkness for 2 h. The absorption of extracts (1.5 mL) was measured at 650 and 665 nm using a spectrophotometer (Evolution 210, Thermo Scientific, Abingdon, UK). The chlorophyll and carotenoid contents were expressed in mg. $g^{-1}$ FW.

### 3.5. Analysis of Heavy Metals Using ICP-MS

The concentrations of Pb, Cd and Cu were analyzed using an ICP-MS instrument for the different parts of the plant using the methodology adopted by Baldi et al. (2021). Six-point calibration curves were used as blanks. A spray chamber and nebulizer were used to introduce the liquid sample into the ICP-MS instrument. The instrument has different heating zones where the sample is successively dried, vaporized, atomized and ionized. In the process, the sample gets transformed from a liquid aerosol to a gas composed of monotonic positively-charged ions and excited atoms.

### 3.6. Phytoextraction Efficiency

Two indices were defined to evaluate the phytoextraction ability of the plants. The bioaccumulation factor (BAF) was calculated thus:

$$\mathrm{BAF} = \frac{\text{Metal Concentration in roots}}{\text{Metal Concentration in the substrate}} \tag{1}$$

The translocation factor (TF) was evaluated for different parts of plants using the following equations:

$$\mathrm{TF_{veg}} = \frac{\text{Concentration in aerial part} - \text{4th mature leaves}}{\text{Concentration in the roots}} \tag{2}$$

$$\mathrm{TF_{shoot,f}} = \frac{\text{Concentration in leaves and stems, at full flowering stage}}{\text{Concentration in the roots}} \tag{3}$$

$$\mathrm{TF_{flower}} = \frac{\text{Concentration in flowers at the full flowering stage}}{\text{Concentration in the roots}} \tag{4}$$

$$\mathrm{TF_{shoot,p}} = \frac{\text{Concentration in leaves and stems measured when pods are mature}}{\text{Concentration in the roots}} \tag{5}$$

$$\mathrm{TF_{pods}} = \frac{\text{Concentration in the mature pods}}{\text{Concentration in the roots}} \tag{6}$$

### 3.7. Statistical Analysis

The experimental design was completely randomized with three replications. ANOVA tests at a level of 5% were carried out to analyze the data obtained for the three HMs investigated for two types of legume crops. The averages were compared using the Duncan multiples range test. The statistical analysis was done using SAS software V9 (SAS Institute, Cary, NC, USA). The variations in heavy metal amounts in the collected plants, as well as their bioaccumulation factors and translocation factors, were tested by one way ANOVA. Moreover, two ways ANOVA was performed to test the variations in plant species, plant organs and their interaction. Pearson correlation analysis was performed to test the linear dependence among the analyzed heavy metals and plant parameters.

## 4. Discussion

In this experimental analysis, the authors focused on two varieties of bean in Tunisia—the common bean and the faba bean, and embarked on investigating the effect of three HMs—Pb, Cu and Cd—on a range of plant-growth aspects, at different stages of plant growth. The addition of HMs to the soil had a detrimental effect on plant growth, with the RFW and SFW being lower vis-à-vis the uncontaminated control samples, in accordance with the results published in [34] for the effects of toxic metals on Arabidopsis. Baldi et al. [16] have noted that the root growth of common bean significantly decreased ($p < 0.05$) with the increase in soil Pb concentration. In this study, the authors considered two other HMs—Cd and Cu—and showed that the adverse growth-inhibitory effects varied among the HMs [34] (Figures 1 and 2). Cadmium turned out to have the most negative effect and copper the least. The negative effect could be due to the suppression of the

elongation growth rate of cells, resulting in an irreversible inhibition exerted by Cd on the proton pump responsible for the process [35].

Additionally, the exposure of the two varieties of beans to these three HMs resulted in a reduction in chlorophyll and carotenoid contents (Table 2). Baek et al. [34] have observed that abiotic stresses—which includes inhibition by toxic metals—on plants tend to affect their chlorophyll biosynthesis capability. Cd and Pb tend to inhibit enzymatic activity of δ-aminolevulinic acid dehydratase and protochlorophyllide reductase associated with chlorophyll biosynthesis. Copper, being an essential micronutrient, was found to be less toxic than the other two HMs, as expected [36]. If one would compare common bean with faba bean, the former was affected more than the latter [37].

However, Zhang et al. [38] have reported that HMs can either enhance or decrease carotenoid production, and that would depend on the type of HM. Carotenoids protect chlorophyll pigments under heavy-metal-induced stress conditions [39,40], and thus if their biosynthesis is disturbed, it would directly hurt the chlorophyll content in the leaves of plants [41,42].

Bioaccumulation and translocation of the HMs were also studied in this experiment. In addition to the quantification of the accumulation of these HMs on roots, shoots, flowers and pods, the BAFs and TFs were also calculated. Roots accumulate a lot of HMs (in this case, Cd and Pb especially), when the plants are in the vegetative stages of growth (4th mature leaf stage) [43]. While the point of entry for the HMs—from the soil to the roots—is the apoplast of the root-cortex and the endoderm [44,45], the upward transport from thereto the stem, branches, leaves, flowers and fruits is termed as translocation. This may or may not require complexing agents such as organic acids, amino acids and peptides [46]. Suman et al. [47] are of the view that metallic trace elements often preferentially accumulate in roots (as Pb does in the case of this study). The low TF values for lead are corroborated by what has been reported by [48,49]. Lead is tightly retained by the root tissues. Faba bean, which has shown higher TF values in general for all the three HMs studied, can be posited as an effective phytoextracting agent in rhizofiltration facilities [8]. However, post-extraction, they will not be available as food commodities. While HMs prefer in general to make the roots their permanent abodes', Aboulroos et al. [50] have reported that they prefer the leaves next and the stem thereafter.

The variation of the HM concentrations for common bean and faba bean at the 4th leaf stage can be explained by the sorption phenomenon exhibited by the HMs in the substrate. The exchanges between the liquid and solid phases of the soil depend on the sorption mechanism referred to. After being adsorbed on the surface of soil particles, the HMs can be absorbed by the mineral particles. A TF value for Pb, Cu and Cd less than unity (Table 4) in the 4th leaf stage, for both the varieties, suggests that the HMs are not effectively transferred from the roots to the above-ground parts of the plants [50].

The results obtained in this study concur with those of [51] who showed that with a Pb concentration in the range of 9 to 267 μM for *Arabispaniculata Franch*, the TF ratio was less than unity. A TF value greater than one was reported for *Solanumnigrum* by [51], suggesting that this plant species can easily transfer lead from the soil to airborne organisms. To reiterate, hyper-accumulators are plants which are effective in moving HMs up from roots to the stems and leaves, and thereby have TF ratios greater than unity [52].

In the vegetative stage, as mentioned earlier too, HMs (lead especially) tend to accumulate in the roots, and thus the two varieties of beans considered in this particular study cannot be hyper-accumulators of lead in this stage of growth. However, when it comes to copper, the TF is close to unity, indicating that copper is transferred from the soil to the roots to the aerial parts. This, as mentioned on a few occasions earlier in this article, is because copper is an essential micronutrient for plant growth, especially in the vegetative stage. The TF values for Cd obtained in this study are lower than those for copper in the vegetative stage. Zhao et al. [53], however, have reported higher TF ratios for Cd in the vegetative stage.

For a plant to be an effective phytoremediating agent, its BAF must be greater than unity [54]. In this study, the BAFs for Pb, Cu and Cd in the vegetative stage of common bean and faba bean were 0.43, 0.81, 0.12, and 0.15, 0.94, 0.14, respectively. One can conclude from these values that *Vicia faba* cannot be an effective phytoremediating agent in the vegetative stage [55].

Most HMs are insoluble in the vascular system of plants, and are thereby less mobile. This is owing to their presence as sulphate, phosphate or carbonates, which immobilises them in the apoplastic (extracellular) and symplastic compartments of the vascular system [56]. The WHO proposes limits of 10 mg/kg biomass and 0.3 mg/kg biomass for Pb and Cd, respectively. As can be seen in Table 4, the Pb and Cd concentrations in the roots, stems, leaves and flowers recorded in this study are above this tolerable upper limit prescribed by the WHO [8].

When it comes to copper, its uptake by plants depends on the ability of plants to take it up through the soil–root interface as a divalent cation or chelated copper, and also on the total amount of Cu present in the soil. Faba bean, as this study showed, is a hyper-accumulator of copper. As reported by Memon et al. [57], if there are free metals available for translocation from roots to shoots, it indicates that the sequestration of these metals in root vacuoles is limited. Copper, though necessary at low doses, is often present in soils at concentrations which may be quite toxic to plants growing there. Values of 0.4 to 45.8 mg Cu per kg of soil have been reported for uncontaminated soils, in [58]. Plants which are tolerant to high concentrations of toxic heavy metals are usually capable of compartmentalizing metal ions, sequestering them in vacuoles and thus excluding them from cell-sites where processes such as cell division and respiration occur [58].

Singh et al. [59] reported that the process of translocating HMs in plant species is a crucial factor in determining the distribution of these elements in different plant tissues. The HMs are mobilized and removed by the root cells, bound by the cell wall and then transported through the plasma membrane, driven by proton pumps dependent on ATP. In addition to cationic nutrients, plant transporters also participate in the migration of potentially toxic cations between plant membranes [60]. This study yielded a TF value of less than unity for lead in the full-flowering stage, indicating a propensity for lead to lodge itself in the roots preferentially, vis-à-vis Cu and Cd. This can lead to the conclusion that the common bean and faba bean are not hyper-accumulators for lead in the vegetative stage. The roots of leguminous plants such as common bean and faba bean secrete exudates necessary for the formation of nodules that facilitate the mobilization and translocation of metals. With TFs for Cu greater than unity in the flowering stage, faba bean plants are hyper-accumulators for the red metal, but just accumulators for Cd (the TF value being less than one). Farmers can thus avail themselves of common bean and faba bean crops as phytoremediating agents to cleanse the soil of Cd (by locking it up in the roots). After removing these crops once enough Cd has been extracted, the soil is much less contaminated then (at least with respect to Cd). In the flowering stage, these two crops are hyper-accumulators of Pb and Cd (BAFs in the range of 1.08 to 2.42).

In the fruiting stage, the stems, leaves and pods of the plants accumulated more Cu than Pb and Cd, for the simple reason that Cu is a catalyst in electron transfer and redox reactions involving enzymatic reactions and protein synthesis [60]. Environmental factors, plant types and the stage of growth of the plant influence the bioaccumulation of HMs. As already discussed before, in very high concentrations, even the essential micronutrients such as copper can be toxic to plants. In the pods stage, both common bean and faba bean are capable of taking up Pb, Cu and Cd, and subsequently sequestering them in the vacuoles of the root system. As the TFs for Cu and Cd are greater than unity in this stage of plant growth for both the varieties, they are hyper-accumulators for these two HMs in the pods-stage, as also concluded by [31] for the Brassica species of plants.

## 5. Conclusions

The effect of HMs in soil is gradually increasing due to the intensification of agriculture, the excessive use of fertilizers and pesticides and especially the use of wastewater in several areas of the country which contributes significantly to the reduction of soil fertility and the reduction of plant growth and productivity. The results of our study suggest the feasibility of classifying faba bean and common bean plants into accumulators and hyper-accumulators according to growth and development stage: during the vegetative growth stage, bean and faba bean plants are classified as accumulators of lead, copper and cadmium ($TF < 1$). These results suggest at this stage plants cannot transfer metal to aerial organs. During the flowering stage, faba bean plants are classified as hyper-accumulators of copper ($TF_s = 1.38$ and $TF_{Fl} = 1.48$), whereas during pod stage, common beans are classified as hyper-accumulators of cadmium ($TF_s = 3.05$ and $TF_g = 3.20$).

Most vegetable species are heavy metal-sensitive at all stages of their lifecycle. Thus, the valorization of legumes in several types of soils could help farmers to overcome the depressive effects of these metals. Lead, copper and cadmium have significant effects on photosynthetic pigments (chlorophyll and carotenoids).

**Author Contributions:** Conceptualization, N.T. and W.S.; methodology, W.S. and N.T.; formal analysis, W.S. and N.T.; investigation, W.S. and N.T.; data curation, W.S. and N.T.; writing—original draft preparation, W.S. and N.T.; writing—review and editing, W.S.; A.M and H.D.-M.; visualization, N.T., K.A. and A.M.; supervision, N.T.; project administration, N.T. and H.D.-M.; funding acquisition, H.D.-M., H.G.-R. and C.S.-G. All authors have read and agreed to the published version of the manuscript.

**Funding:** The paper is funded by the Universidad Autónoma de Zacatecas unidad Académica de Ingeniería Eléctrica Jdn. Juárez. This research was also jointly founded by National research laboratory LR21AGR05.

**Institutional Review Board Statement:** Not applicable.

**Informed Consent Statement:** Not applicable.

**Data Availability Statement:** Not applicable.

**Conflicts of Interest:** The authors declare no conflict of interest. The funders had no role in the design of the study; in the collection, analyses, or interpretation of data; in the writing of the manuscript; or in the decision to publish the results.

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
