# Peer review of "Effects of Lead, Copper and Cadmium on Bioaccumulation and Translocation Factors and Biosynthesis of Photosynthetic Pigments in Vicia faba L. (Broad Beans) at Different Stages of Growth"

_applsci, doi:10.3390/app12188941_

Round 1

Reviewer 1 Report

The topic of this paper has significant importance, the resulting data are also very interesting, but the manner of presentation, especially their interpretation is deficient. I suggest a major revision trying to use a more scientific way of interpretation, avoiding such interpretations ", RC>SC>PC for Pb, SC>RC>PC for Cu, and PC>RC>SC for Cd"; " as tabulated", "with respect to the control", etc., using an impersonal mode, formulating clear and concise conclusions based exclusively on the results obtained.

Author Response

Response to Reviewer 1 Comments

Point 1: The topic of this paper has significant importance, the resulting data are also very interesting, but the manner of presentation, especially their interpretation is deficient. I suggest a major revision trying to use a more scientific way of interpretation, avoiding such interpretations ", RC>SC>PC for Pb, SC>RC>PC for Cu, and PC>RC>SC for Cd"; " as tabulated", "with respect to the control", etc., using an impersonal mode, formulating clear and concise conclusions based exclusively on the results obtained.

Response 1: I adhered to the major correction: I replace RC>SC>PC for Pb, SC>RC>PC for Cu, and  PC>RC>SC for Cd" by roots accumulate more metals than shoots and pods for Pb, whereas, for Cu, shoots accumulate more metals than roots and pods and for Cd, pods accumulate more metals than roots and shoots. For faba bean, in the case of Pb and Cd, roots recorded the highest concentration”.

I replace "with respect to the control" by compared to and I change some sentences in results.

I change conclusion, and the new one is below:

  1. Conclusions

The effect of HMs in soil is gradually increasing due to the intensification of agriculture, the excessive use of fertilizers and pesticides and especially the use of wastewater treated in several areas of the country which contributes significantly to the reduction of soil fertility and the reduction of plant growth and productivity. Most vegetable species are heavy metals-sensitive at all stages of their lifecycle. Thus, the valorization of legumes in several types of soils could help farmers to overcome the depressive effects of these metals. The results of our study suggest the feasibility of classifying faba bean and common bean plants into accumulators and hyper-accumulators according to growth and development stage: during the vegetative growth stage, bean and faba bean plants are classified as accumulators of lead, copper and cadmium (TF < 1), these results suggest at this stage plants con not transfer metal to aerial organs. During the flowering stage, faba bean plants are classified as hyper-accumulator of copper (TFs = 1.38 and TFFl = 1.48), whereas, during pods stage common bean are classified as hyper-accumulator of cadmium (TFs = 3.05 and TFg = 3.20). Lead, copper and cadmium have significant effects on photosynthetic pigments (chlorophyll and carotenoids). Such an approach may considerably upgrade all procedures aimed at selecting heavy metals-tolerant varieties to be exploited for cultivation in contaminated soil. Alternative options should be carried out in order to prevent excessive accumulation of heavy metals and all vegetables should be washed properly before consumption as washing can remove a significant amount of aerial contamination from the vegetable surface.

Reviewer 2 Report

Dear Authors,

Generally, the manuscript is very well written, the data is obviously interpreted and the figures are well presented.

But it is important to list the toxicity limits to know how far the research held from these limits.

Toxicity limits of: the Plant, in the soil, for man, for animal for each element in the study is needed. 

This manuscript does not undergo the special issue submitted to it, so please revise this issue needs.

wish you the best

Author Response

Response to Reviewer 2 Comments

Point 1: Generally, the manuscript is very well written, the data is obviously interpreted and the figures are well presented. But it is important to list the toxicity limits to know how far the research held from these limits. Toxicity limits of: the Plant, in the soil, for man, for animal for each element in the study is needed. 

This manuscript does not undergo the special issue submitted to it, so please revise this issue needs.

 Response 1: I adhered to the major correction: I add toxicity limits:

Heavy metals are significant environmental pollutants, and their toxicity is a problem of increasing significance for ecological, evolutionary, nutritional and environmental reasons. Environment is defined as totally circumstances surrounding an organism or group of organisms especially, the combination of external physical conditions that affect and influence the growth, development and survival of organisms [11]. A pollutant is any substance in the environment, which causes objectionable effects, impairing the welfare of the environment, reducing the quality of life and may eventually cause death. Such a substance has to be present in the environment beyond a set or tolerance limit. Hence, environmental pollution is the presence of a pollutant in the environment air, water and soil, which may be poisonous or toxic and will cause harm to living things in the polluted environment.

The regulatory limit of cadmium (Cd) in agricultural soil is 100 mg/kg soil. Plants grown in soil containing high levels of Cd show visible symptoms of injury reflected in terms of chlorosis, growth inhibition, browning of root tips and finally death [18]. Cd has been shown to interfere with the uptake, transport and use of several elements (Ca, Mg, P and K) and water by plants [12]. Cd also reduced the absorption of nitrate and its transport from roots to shoots, by inhibiting the nitrate reductase activity in the shoots [19]. Copper (Cu) is considered as a micronutrient for plants and plays important role in CO2 assimilation and ATP synthesis. Cu is also an essential component of various proteins like plastocyanin of photosynthetic system and cytochrome oxidase of respiratory electron transport chain [13], but excess of Cu in soil plays a cytotoxic role, induces stress and causes injury to plants. This leads to plant growth retardation and leaf chlorosis. Exposure of plants to excess Cu generates oxidative stress and ROS [14]. Oxidative stress causes disturbance of metabolic pathways and damage to macromolecules [15].

Reviewer 3 Report

Dear Authors,

As a Reviewer, I have a few comments about your manuscript, namely:

1. Research questions should appear in the summary.

2. At the end of the chapter: Introduction "research hypotheses should be formulated.

3. The chapter: Research methodology "should be before the discussion of the research results, and not at the very end of the manuscript. Moreover, the methodology should develop the calculations of correlation and regression coefficients and describe them in the chapter" Research results ", and comment in the chapter" discussion ".

4. Chapter conclusions should be formatted, i.e. the bullets should be removed. Present the concluded issues in separate paragraphs, but summarize them in more detail, referring to the introduced research hypotheses and research questions.

5. Literature - items older than 2000, should be replaced with newer items, i.e. items 33, 47.

After considering the above-mentioned comments, the article may be published.

Author Response

Response to Reviewer 3 Comments

Point 1:  Research questions should appear in the summary.

Point 2:  At the end of the chapter: Introduction "research hypotheses should be formulated.

Point 3: The chapter: Research methodology "should be before the discussion of the research results, and not at the very end of the manuscript. Moreover, the methodology should develop the calculations of correlation and regression coefficients and describe them in the chapter" Research results ", and comment in the chapter" discussion ".

Point 4: Chapter conclusions should be formatted, i.e. the bullets should be removed. Present the concluded issues in separate paragraphs, but summarize them in more detail, referring to the introduced research hypotheses and research questions.

Point 5:  Literature - items older than 2000, should be replaced with newer items, i.e. items 33, 47.

After considering the above-mentioned comments, the article may be published.

Response 1: I put the research question in the summary:

Abstract: Trace elements in the environmental media contribute to toxicities of different types.  Their presence in the arable pedosphere is a human-health risk factor. This study focused on Vicia faba represented by two Tunisian varieties of bean (Mamdouh) and faba bean (Badii). The objective was to analyze the effects of lead, copper and cadmium on their growth, chlorophyll-content and carotenoids-content, as well as the bioaccumulation and translocation factor, at different stages of growth. For each metal, the concentrations the plants were subjected to were 6, 0.3 and 0.03 mg/L of the metal in the compound for lead nitrate, copper nitrate and cadmium acetate, respectively. The analysis was carried out using an atomic absorption spectrophotometer (ICP-MS), encompassing all the parts of the plant. The authors detected a perceptible decrease in the fresh weight of roots and shoots, as well as a drop in the chlorophyll and carotenoid contents, for all the three heavy metals. Cadmium turned out to be the most toxic of the three metals and copper (which is incidentally an essential micronutrient for plant growth) the least. As far as the bioaccumulation factor was concerned, bean and faba bean exhibited different behaviours, both with regard to the growth stages and the heavy metal absorbed. During the vegetative growth stage, both were accumulators of all the three heavy metals (a translocation factor less than unity). However, in the flowering stage, faba bean was a hyper-accumulator of copper (TF > 1); while the bean plants accumulated a lot of lead in the pods-stage (TF>1). It is worthwhile to pose new research questions and try to answer them in this study, if legumes are accumulator or hyper accumulator plants in which stage and in where organ accumulate more HMs.

Response 2: At the end of the chapter: Introduction "research hypotheses were formulated:

In Tunisia, soil polluted with heavy metals is gradually increasing due to the scarcity of rains and the use of recycled wastewater for irrigation as is the case of semiaride regions. In legumes, heavy metals causes various physiological and biochemical alterations and diverse toxicity symptoms such as chlorosis and necrosis [32]. Despite the acquired knowledge in relation to the response of species to heavy metals, there is a gap in relevant research fields for grai legumes. Considering that grain legumes as accumulator plants for some heavy metals, this study aimed at investigating the response of two locale varieties of grain legumes to lead, copper and cadmium at different growth stage. As such, two Tunisian varieties of common bean and faba bean were subjected to different concentrations of lead, copper and cadmiu at development stage, flowering stage and pods stage, and their response was assessed on the basis of the  translocation factor (TF) and the biochemical responses of the crops (chlorophyll and carotenoid contents).

Response 3: I change the chapter: Research methodology before the discussion and I mentioned correlation and interaction in tne methodology.

Response 4: Conclusion was formulated:

The effect of HMs in soil is gradually increasing due to the intensification of agriculture, the excessive use of fertilizers and pesticides and especially the use of wastewater treated in several areas of the country which contributes significantly to the reduction of soil fertility and the reduction of plant growth and productivity. Most vegetable species are heavy metals-sensitive at all stages of their lifecycle. Thus, the valorization of legumes in several types of soils could help farmers to overcome the depressive effects of these metals. The results of our study suggest the feasibility of classifying faba bean and common bean plants into accumulators and hyper-accumulators according to growth and development stage: during the vegetative growth stage, bean and faba bean plants are classified as accumulators of lead, copper and cadmium (TF < 1), these results suggest at this stage plants con not transfer metal to aerial organs. During the flowering stage, faba bean plants are classified as hyper-accumulator of copper (TFs = 1.38 and TFFl = 1.48), whereas, during pods stage common bean are classified as hyper-accumulator of cadmium (TFs = 3.05 and TFg = 3.20). Lead, copper and cadmium have significant effects on photosynthetic pigments (chlorophyll and carotenoids). Such an approach may considerably upgrade all procedures aimed at selecting heavy metals-tolerant varieties to be exploited for cultivation in contaminated soil. Alternative options should be carried out in order to prevent excessive accumulation of heavy metals and all vegetables should be washed properly before consumption as washing can remove a significant amount of aerial contamination from the vegetable surface.

Response 5: Items 33 and 47 were replaced.

Round 2

Reviewer 1 Report

- line 29 from abstract- delete the content from sentences

-line 38 - too many keywords

- line 43 -add Zn in the sentences

- line 137- correct "grai"

- line 150- delete "Refer"

- line 152- 153- "If one would..." this must be part of the discussion section

- line 154 - delete "for the shoots (leaves and stems)"

 -line 176- "The concentration of HMs are naturally higher in the roots..." please delete these or add references because you don't determine the concentration of the HMs from the roots

-lines 180,  202, 213 - please avoid using "drastic" or 'spectacular", "striking" or " embarked" - these are not specific to the research field

- line 203- the sentence is not finished

- line 211- please change "The data gathered"

- line 235- "A significant difference vis-à-vis the controls was seen"-  for what? it is not clear... please complete your affirmation

- line 313- please change "tabulated"

- line 322- at Treatment applied section- please be more specifically: "For each metal the concentration were 6, 0.3 and 0.03... or for lead the concentration was 6 mg/L; for copper was 0.3, and for Cadmium was 0.03

- at the Sample collection Section- please be more specific about the growth stages, eventually, you can use the Fehr and Caviness stages from soybean if is not a specific scale for bean

- line 333- please delete "To obviate additional..."- it is clear that in the research field must be worked meticulously

- line 341- please change "at the fully vegetative stages" with the analyzed stage (anthesis or another stage)

- line 371- please delete "moment"

- line 372- correct "testr"

- line 390- attention to the citation of the authors

- line 396- please reformulate

- in the conclusion section- please add the conclusion based on your research, so lines 494-499 can be moved to the introduction section

Author Response

Response to Reviewer 2

Comments

Point 1  line 29 from abstract- delete the content from sentences

Point 2  line 38 - too many keywords

Point 3 line 43 -add Zn in the sentences

Point 4 line 137- correct "grai"

Point 5 line 150- delete "Refer"

Point 6  line 152- 153- "If one would..." this must be part of the discussion section

Point 7 line 154 - delete "for the shoots (leaves and stems)"

 Point 8 line 176- "The concentration of HMs are naturally higher in the roots..." please delete these or add references because you don't determine the concentration of the HMs from the roots

Point 9 lines 180,  202, 213 - please avoid using "drastic" or 'spectacular", "striking" or " embarked" - these are not specific to the research field

Point 10  ine 203- the sentence is not finished

Point 11 line 211- please change "The data gathered"

Point 12 line 235- "A significant difference vis-à-vis the controls was seen"-  for what? it is not clear... please complete your affirmation

Point 13 line 313- please change "tabulated"

Point 14 line 322- at Treatment applied section- please be more specifically: "For each metal the concentration were 6, 0.3 and 0.03... or for lead the concentration was 6 mg/L; for copper was 0.3, and for Cadmium was 0.03

Point 15 at the Sample collection Section- please be more specific about the growth stages, eventually, you can use the Fehr and Caviness stages from soybean if is not a specific scale for bean

Point 16 line 333- please delete "To obviate additional..."- it is clear that in the research field must be worked meticulously

Point 17 line 341- please change "at the fully vegetative stages" with the analyzed stage (anthesis or another stage)

Point 18 line 371- please delete "moment"

Point 19 line 372- correct "testr"

Point 20 line 390- attention to the citation of the authors

Point 21 ine 396- please reformulate

Point 22 in the conclusion section- please add the conclusion based on your research, so lines 494-499 can be moved to the introduction section

Responses

Response 1 : « Content » was deleted

Response 2 : keywords were reduced

Response 3 : Zn was added

Response 4 : grai corrected with grain

Response 5 : Refer was deleted

Response 6 : line 152-153 were added in discussion section

Response 7 : "for the shoots (leaves and stems)" was deleted

Response 8 : "The concentration of HMs are naturally higher in the roots..." was deleted

Response 9 : drastic" or 'spectacular", "striking" or " embarked were replaced by strong

Response 10 : Line 203- the sentence was finished

Response 11 : "The data gathered" was replaced by the data collected

Response 12 : line 235 was affirmed

Response 13 : "tabulated" was replaced by showed

Response 14 : For lead the concentration was 6 mg/L, for copper was 0.3 mg/L and for cadmium was 0.03 mg/L

Response 15 : At different different stages of growth (vegetative stage and reproductive stage)

Response 16 : "To obviate additional..." was deleted

Response 17 : "at the fully vegetative stages" was replaced by at anthesis stage

Response 18 : "moment" was deleted

Response 19 : "testr" was corrected by test

Response 20 : Citations were changed

Response 21 : Regarding carotenoids, this study recorded decreases when plants were exposed to HMs

Response 22 : lines 494-499 were moved to the introduction section

Reviewer 2 Report

Dear Authors,

Thanks for finalizing this manuscript. Wish you the best

Author Response

No Comments
